# INCREMENTAL RANDOMIZED SMOOTHING CERTIFICATION

**Shubham Ugare**[1], **Tarun Suresh**[1], **Debangshu Banerjee**[1], **Gagandeep Singh**[1,2], **Sasa Misailovic**[1]

[1]University of Illinois Urbana-Champaign,  [2]VMware Research

{sugare2, tsuresh3, db21, ggnds, misailo}@illinois.edu

## ABSTRACT

Randomized smoothing-based certification is an effective approach for obtaining robustness certificates of deep neural networks (DNNs) against adversarial attacks. This method constructs a smoothed DNN model and certifies its robustness through statistical sampling, but it is computationally expensive, especially when certifying with a large number of samples. Furthermore, when the smoothed model is modified (e.g., quantized or pruned), certification guarantees may not hold for the modified DNN, and recertifying from scratch can be prohibitively expensive.

We present the first approach for incremental robustness certification for randomized smoothing, IRS. We show how to reuse the certification guarantees for the original smoothed model to certify an approximated model with very few samples. IRS significantly reduces the computational cost of certifying modified DNNs while maintaining strong robustness guarantees. We experimentally demonstrate the effectiveness of our approach, showing up to 4.1x certification speedup over the certification that applies randomized smoothing of approximate model from scratch.

## 1 INTRODUCTION

Ensuring the robustness of deep neural networks (DNNs) to input perturbations is gaining increased attention from both users and regulators in various application domains (Bojarski et al., 2016; Amato et al., 2013; Julian et al., 2018; ISO). Out of many techniques for obtaining robustness certificaties, statistical methods currently offer the greatest scalability. Randomized smoothing (RS) is a popular statistical certification method by constructing a smoothed model $g$ from a base network $f$ under noise (Cohen et al., 2019). To certify the model $g$ on an input, RS certification checks if the estimated lower bound on the probability of the top class is greater than the upper bound on the probability of the runner-up class (with high confidence). RS certification computes the certified accuracy metric of the DNN on the set of test inputs as a proxy for the DNN robustness. However, despite its effectiveness, RS-based certification can be computationally expensive as it requires DNN inference on a large number of corruptions per input.

The high cost of certification complicates the DNN deployment process, which has become increasingly iterative: the networks are often modified post-training to improve their execution time and/or accuracy. Especially, deploying DNNs on real-world systems with bounded computing resources (e.g., edge devices or GPUs with limited memory), has led to various techniques for approximating DNNs. Common approximation techniques include quantization – reducing the numerical precision of weights (Fiesler et al., 1990; Balzer et al., 1991), and pruning – removing weights that have minimal impact on accuracy (Janowsky, 1989; Reed, 1993).

Common to all of these approximations is that the network behavior (e.g., the classifications) remains the same on most inputs, its architecture does not change, and many weights are only slightly changed. When a user seeks to select a robust and accurate DNN from these possible approximations, RS needs to be performed to compute the robustness of all candidate networks. For instance, in the context of approximation tuning, there are multiple choices for approximation where different quantization or pruning strategies are applied at different layers. Tools such as (Chen et al., 2018b;a; Sharif et al., 2019; Zhao et al., 2023) use approximations iteratively and test the network at each step. To ensure DNN robustness when using such tools, one would need to check certified accuracy, computed using RS on test data in each step. However, performing RS to compute certified accuracy from scratch can take hours as shown in our experiments even for a single network (with only 500 test images).

Therefore, a major encumbrance of almost all existing RS-based certification practices in the above setting, is that *the expensive certification needs to be re-run from scratch* for each approximate network. Overcoming this main limitation requires addressing the following fundamental problem:

> How can we leverage the information generated while certifying a given network to speed up the certification of similar networks?

**This Work.** We present the first incremental RS-based certification framework called Incremental Randomized Smoothing (IRS) to answer this question. The primary objective of our work is to improve the sample complexity of the certification process of similar networks on a predefined test set. Improved sample complexity results in overall speedup in certification, and it reduces the energy requirement and memory footprint of the certification. Given a network $f$ and its smoothed version $g$, and a modified network $f^p$ with its smoothed version $g^p$, IRS incrementally certifies the robustness of $g^p$ by reusing the information from the execution of RS certification on $g$.

IRS optimizes the process of certifying the robustness of smoothed classifier $g^p$ on an input $x$, by estimating *the disparity* $\zeta_x$ – the upper bound on the probability that outputs of $f$ and $f^p$ are distinct. Our new algorithm is based on three key insights about disparity:

1. Common approximations yield small $\zeta_x$ values – for instance, it is below 0.01 for int8 quantization for multiple large networks in our experiments.
2. Estimating $\zeta_x$ through binomial confidence interval requires fewer samples as it is close to 0 – it is, therefore, less expensive to certify with this probability than directly working with lower and upper probability bounds in the original RS algorithm.
3. We can leverage $\zeta_x$ alongside the bounds in the certified radius of $g$ around $x$ to compute the certified radius for $g^p$ – thus soundly reusing the samples from certifying $g$.

We extensively evaluate the performance of IRS when applying several common DNN approximations such as pruning and quantization on state-of-the-art DNNs on CIFAR10 (ResNet-20, ResNet-110) and ImageNet (ResNet-50) datasets.

**Contributions.** The main contributions of this paper are:

- We propose a novel concept of incremental RS certification of the robustness of the updated smoothed classifier by reusing the certification guarantees for the original smoothed classifier.
- We design the first algorithm IRS for incremental RS that efficiently computes the certified radius of the updated smoothed classifier.
- We present an extensive evaluation of the performance of IRS speedups of up to 4.1x over the standard non-incremental RS baseline on state-of-the-art classification models.

IRS code is available at https://github.com/uiuc-arc/Incremental-DNN-Verification.

## 2 BACKGROUND

**Randomized Smoothing.** Let $f : \mathbb{R}^m \to \mathcal{Y}$ be an ordinary classifier. A smoothed classifier $g : \mathbb{R}^m \to \mathcal{Y}$ can be obtained from calculating the most likely result of $f(x+\epsilon)$ where $\epsilon \sim \mathcal{N}(0, \sigma^2 I)$.

$$g(x) := \arg\max_{c \in \mathcal{Y}} \mathbb{P}_\epsilon(f(x + \epsilon) = c)$$

The smoothed network $g$ satisfies following guarantee for a single network $f$:

**Theorem 1.** *[From (Cohen et al., 2019)] Suppose $c_A \in \mathcal{Y}$, $\underline{p_A}, \overline{p_B} \in [0, 1]$. if*

$$\mathbb{P}_\epsilon(f(x + \epsilon) = c_A) \geq \underline{p_A} \geq \overline{p_B} \geq \max_{c \neq c_A} \mathbb{P}_\epsilon(f(x + \epsilon) = c),$$

*then $g(x + \delta) = c_A$ for all $\delta$ satisying $\|\delta\|_2 \leq \frac{\sigma}{2}(\Phi^{-1}(\underline{p_A}) - \Phi^{-1}(\overline{p_B}))$, where $\Phi^{-1}$ denotes the inverse of the standard Gaussian CDF.*

Computing the exact probabilities $P_\epsilon(f(x + \epsilon) = c)$ is generally intractable. Thus, for practical applications, CERTIFY (Cohen et al., 2019) (Algorithm 1) utilizes sampling: First, it takes $n_0$ samples to determine the majority class, then $n$ samples to compute a lower bound $\underline{p_A}$ to the success probability with confidence $1 - \alpha$ via the Clopper-Pearson lemma (Clopper and Pearson, 1934). If $\underline{p_A} > 0.5$, we set $\overline{p_B} = 1 - \underline{p_A}$ and obtain radius $R = \sigma \cdot \epsilon \cdot \Phi^{-1}(\underline{p_A})$ via Theorem 1, else we return ABSTAIN.

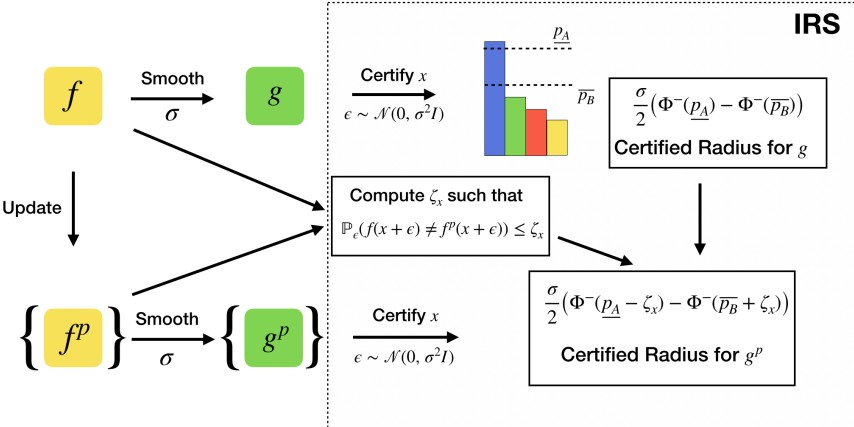

Figure 1: Workflow of IRS from left to right. IRS takes the classifier $f$ and input $x$. IRS reuses the $\underline{p_A}$ and $\overline{p_B}$ estimates computed for $f$ on $x$ by RS. IRS estimate $\zeta_x$ from $f$ and $f^p$. For the smoothed classifier $g^p$ obtained from any of the approximate classifiers $f^p$ it computes the certified radius by combining $\underline{p_A}$ and $\overline{p_B}$ with $\zeta_x$.

**DNN approximation.** DNN weights need to be quantized to the appropriate datatype for deploying them on various edge devices. DNN approximations are used to compress the model size at the time of deployment, to allow inference speedup and energy savings without significant accuracy loss. While IRS can work with most of these approximations, for the evaluation, we focus on quantization and pruning as these are the most common ones (Zhou et al., 2017; Frankle and Carbin, 2019).

---

**Algorithm 1** RS certification (Cohen et al., 2019)

**Inputs:** f: DNN, $\sigma$: standard deviation, $x$: input to the DNN, $n_0$: number of samples to predict the top class, $n$: number of samples for computing $\underline{p_A}$, $\alpha$: confidence parameter

1: **function** CERTIFY($f, \sigma, x, n_0, n, \alpha$)
2:    $counts_0 \leftarrow$ SampleUnderNoise($f, x, n_0, \sigma$)
3:    $\hat{c}_A \leftarrow$ top index in $counts_0$
4:    $counts \leftarrow$ SampleUnderNoise($f, x, n, \sigma$)
5:    $\underline{p_A} \leftarrow$ LowerConfidenceBound($counts[\hat{c}_A], n, 1 - \alpha$)
6:    **if** $\underline{p_A} > \frac{1}{2}$ **then**
7:      **return** prediction $\hat{c}_A$ and radius $\sigma \cdot \Phi^{-1}(\underline{p_A})$
8:    **else**
9:      **return** ABSTAIN

---

## 3    INCREMENTAL RANDOMIZED SMOOTHING

Figure 1 illustrates the high-level idea behind the workings of IRS. It takes as input the classifier $f$, the updated classifier $f^p$, and an input $x$. Let $g$ and $g^p$ denote the smoothed network obtained from $f$ and $f^p$ using RS respectively. IRS reuses the $\underline{p_A}$ and $\overline{p_B}$ estimates computed for $g$ to compute the certified radius for $g^p$.

### 3.1    MOTIVATION

**Insight 1: Similarity in approximate networks** We observe that for many practical approximations, $f$ and $f^p$ produce the same result on most inputs. In this experiment, we estimate the disparity between $f$ and $f^p$ on Gaussian corruptions of the input $x$. We compute a lower confidence bound $\zeta_x$ such that $\mathbb{P}_\epsilon(f(x + \epsilon) \neq f^p(x + \epsilon)) \leq \zeta_x$ for $\epsilon \sim \mathcal{N}(0, \sigma^2 I)$.

Table 1 presents empirical average $\zeta_x$ for int8 quantization and pruning 10% lowest magnitude weights for some of the networks in our experiments computed over 500 inputs. We

Table 1: Average $\zeta_x$ with $n = 1000$ samples for various approximations.

|  | CIFAR10 ResNet-110 | ImageNet ResNet-50 |
|---|---|---|
| int8 | 0.009 | 0.006 |
| prune10 | 0.010 | 0.008 |

compute $\zeta_x$ value as the binomial confidence upper limit using (Clopper and Pearson, 1934) method with $n = 1000$ samples with $\sigma = 1$. The results show that the $\zeta_x$ value is quite small in all the cases.

**Insight 2: Sample reduction through $\zeta_x$ estimation** We demonstrate that $\zeta_x$ estimation for approximate networks is more efficient than running certification from scratch. Fig. 2 shows that for the fixed target error $\chi$, confidence $(1 - \alpha)$ and estimation technique, the number of samples required for estimation peaks, when the actual parameter value is around 0.5 and is smallest around

the boundaries. For example, when $\chi = 0.5\%$ and $\alpha = 0.01$ estimating the unknown binomial proportion will take $41,500$ samples if the actual parameter value is $0.05$ while achieving the same target error and confidence takes $216,900$ samples (5.22x higher) if the actual parameter value is $0.5$. As observed in the previous section, $\zeta_x$'s value for many practical approximations is close to 0.

Leveraging the observation shown in Fig. 2 and given actual value $\zeta_x$ is close to 0, estimating $\zeta_x$ with existing binomial proportion estimation techniques is efficient and requires a smaller number of samples. In Appendix A.7, we show the distribution of $\underline{p_A}$ and $\overline{p_B}$ for various cases. We see that $\underline{p_A}$ and $\overline{p_B}$ do not always lie close to 0 or 1 and have a more dispersed distribution. Thus, estimating those requires more samples. Prior work (Thulin, 2013) has theoretically shown that the expected length of the confidence interval for Clopper-Pearson follows a similar trend as in Fig. 2. This theoretical result supports our observation. We show in Appendix A.1 that this observation is not contingent on a specific estimation method and holds for other popular estimation techniques, e.g., (Wilson, 1927), (Agresti and Coull, 1998).

**Insight 3: Computing the approximate network's certified radius using $\zeta_x$** For certification of the approximate network $g^p$, our main insight is that estimating $\zeta_x$ and using that value to compute the certified radius is more efficient than computing RS certified radius from scratch. The next theorem shows how to use estimated value of $\zeta_x$ to certify $g^p$ (the proof is in Appendix A.2):

**Theorem 2.** *If a classifier $f^p$ is such that for all $x \in \mathbb{R}^m, \mathbb{P}_\epsilon(f(x + \epsilon) \neq f^p(x + \epsilon)) \leq \zeta_x$, and classifier $f$ satisfies $\mathbb{P}_\epsilon(f(x + \epsilon) = c_A) \geq \underline{p_A} \geq \overline{p_B} \geq \max_{c \neq c_A} \mathbb{P}_\epsilon(f(x + \epsilon) = c)$ and $\underline{p_A} - \zeta_x \geq \overline{p_B} + \zeta_x$ then $g^p$ satisfies $g^p(x + \delta) = c_A$ for all $\delta$ satisying $\|\delta\|_2 \leq \frac{\sigma}{2}(\Phi^{-1}(\underline{p_A} - \zeta_x) - \Phi^{-1}(\overline{p_B} + \zeta_x))$*

Theorem 1 considers standard RS for a single network. Our Theorem 2 shows how to use the estimated value of $\zeta_x$ to transfer the certification guarantees across two networks $f$ and $f^p$.

Figure 2: The number of samples for the Clopper-Pearson method to achieve a target error $\chi$ with confidence $(1 - \alpha)$.

## 3.2 IRS CERTIFICATION ALGORITHM

The Algorithm 2 presents the pseudocode for the IRS algorithm, which extends RS certification from Algorithm 1. The algorithm takes the modified classifier $f^p$ and certifies the robustness of $g^p$ around $x$. The input $n_p$ denotes the number of Gaussian corruptions used by the algorithm.

The IRS algorithm utilizes a cache $\mathcal{C}_f$, which stores information obtained from the RS execution of the classifier $f$ for each input $x$. The cached information is crucial for the operation of IRS. $\mathcal{C}_f$ stores the top predicted class index $\hat{c}_A$ and its lower confidence bound $\underline{p_A}$ for $f$ on input $x$.

The standard RS algorithm takes a conservative value of $\overline{p_B}$ by letting $\overline{p_B} = 1 - \underline{p_A}$. This works reasonably well in practice and simplifies the computation of certified radius $\frac{\sigma}{2}(\Phi^{-1}(\underline{p_A}) - \Phi^{-1}(\overline{p_B}))$ to $\sigma\Phi^{-1}(\underline{p_A})$. We make a similar choice in IRS, which simplifies the certified radius calculation from $\frac{\sigma}{2}(\Phi^{-1}(\underline{p_A} - \zeta_x) - \Phi^{-1}(\overline{p_B} + \zeta_x))$ of Theorem 2 to $\sigma\Phi^{-1}(\underline{p_A} - \zeta_x)$ as we state in the next theorem (the proof is in Appendix A.2):

---

**Algorithm 2** IRS algorithm: Certification with cache

**Inputs:** $f^p$: DNN obtained from approximating $f$, $\sigma$: standard deviation, $x$: input to the DNN, $n_p$: number of Gaussian samples used for certification, $\mathcal{C}_f$: stores the information to be reused from certification of $f$, $\alpha$ and $\alpha_\zeta$: confidence parameters, $\gamma$: threshold hyperparameter to switch between estimation methods

1: **function** CERTIFYIRS($f^p, \sigma, x, n_p, \mathcal{C}_f, \alpha, \alpha_\zeta, \gamma$)
2:    $\hat{c}_A \leftarrow$ top index in $\mathcal{C}_f[x]$
3:    $\underline{p_A} \leftarrow$ lower confidence of $f$ from $\mathcal{C}_f[x]$
4:    **if** $\underline{p_A} < \gamma$ **then**
5:      $\zeta_x \leftarrow$ EstimateZeta($f^p, \sigma, x, n_p, \mathcal{C}_f, \alpha_\zeta$)
6:      **if** $\underline{p_A} - \zeta_x > \frac{1}{2}$ **then**
7:        **return** prediction $\hat{c}_A$ and radius $\sigma\Phi^{-1}(\underline{p_A} - \zeta_x)$
8:    **else**
9:      $counts \leftarrow$ SampleUnderNoise($f^p, x, n_p, \sigma$)
10:      $p'_A \leftarrow$ LowerConfidenceBound(
11:           $counts[\hat{c}_A], n_p, 1 - (\alpha + \alpha_\zeta))$
12:      **if** $p'_A > \frac{1}{2}$ **then**
13:        **return** prediction $\hat{c}_A$ and radius $\sigma\Phi^{-1}(p'_A)$
14:    **return** ABSTAIN

---

**Theorem 3.** *If $\underline{p_A} - \zeta_x \geq \frac{1}{2}$, then $\sigma\Phi^{-1}(\underline{p_A} - \zeta_x) \leq \frac{\sigma}{2}(\Phi^{-1}(\underline{p_A} - \zeta_x) - \Phi^{-1}(\overline{p_B} + \zeta_x))$*

As per our insight 2 (Section 3.1), binomial confidence interval estimation requires fewer samples for binomial with actual probability close to 0 or 1. IRS can take advtange of this when $\underline{p_A}$ is not close to 1. However, when $\underline{p_A}$ is close to 1 then there is no benefit of using $\zeta_x$-based certified radius for $g^p$. Therefore, the algorithm uses a threshold hyperparameter $\gamma$ close to 1 that is used to switch between certified radius from Theorem 2 and standard RS from Theorem 1.

If the $\underline{p_A}$ is less than the threshold $\gamma$, then an estimate of $\zeta_x$ for classifier $f^p$ and the classifier $f$ is computed using the EstimateZeta function. We discuss EstimateZeta procedure in the next section. If the $\underline{p_A} - \zeta_x$ is greater than $\frac{1}{2}$, then the top predicted class in the cache is returned as the prediction with the radius $\sigma\Phi^{-1}(\underline{p_A} - \zeta_x)$ as computed in Theorem 3.

In case, $\underline{p_A}$ is greater than the threshold $\gamma$, similar to standard RS, the IRS algorithm draws $n^p$ samples of $f^p(x + \epsilon)$ by running $n^p$ noise-corrupted copies of $x$ through the classifier $f^p$. The function SampleUnderNoise($f^p, x, n_p, \sigma$) in the pseudocode draws $n_p$ samples of noise, $\epsilon_1 \ldots \epsilon_{n_p} \sim \mathcal{N}(0, \sigma^2 I)$, runs each $x + \epsilon_i$ through the classifier $f^p$, and returns a vector of class counts. If the lower confidence bound is greater than $\frac{1}{2}$, the top predicted class is returned as the prediction with a radius based on the lower confidence bound $\underline{p_A}$.

If the function does certify the input in both of the above cases, it returns ABSTAIN.

The hyperparameters $\alpha$ and $\alpha_\zeta$ denote confidence of IRS results. The IRS algorithm result is correct with confidence at least $1 - (\alpha + \alpha_\zeta)$. For the case $\underline{p_A} \geq \gamma$, this holds since we follow the same steps as standard RS. The function LowerConfidenceBound($counts[\hat{c}_A], n_p, 1 - (\alpha + \alpha_\zeta)$) in the pseudocode returns a one-sided $1 - (\alpha + \alpha_\zeta)$ lower confidence interval for the Binomial parameter $p$ given a sample $counts[\hat{c}_A] \sim Binomial(n_p, p)$. We next state the theorem that shows the confidence of IRS results in the other case when $\underline{p_A} < \gamma$ (the proof is in Appendix A.2):

**Theorem 4.** *If $\mathbb{P}_\epsilon(f(x + \epsilon) = f^p(x + \epsilon)) > 1 - \zeta_x$ with confidence at least $1 - \alpha_\zeta$. If classifier $f$ satisfies $\mathbb{P}_\epsilon(f(x + \epsilon) = c_A) \geq \underline{p_A}$ with confidence at least $1 - \alpha$. Then for classifier $f^p$, $\mathbb{P}_\epsilon(f^p(x + \epsilon) = c_A) \geq \underline{p_A} - \zeta_x$ with confidence at least $1 - (\alpha + \alpha_\zeta)$*

### 3.3 ESTIMATING THE UPPER CONFIDENCE BOUND $\zeta_x$

In this section, we present our method for estimating $\zeta_x$ such that $\mathbb{P}_\epsilon(f(x + \epsilon) \neq f^p(x + \epsilon)) \leq \zeta_x$ with high confidence (Algorithm 3). We use the Clopper-Pearson (Clopper and Pearson, 1934) method to estimate the upper confidence bound $\zeta_x$.

We store the *seeds* used for randomly generating Gaussian samples while certifying the function $f$ in the cache, and we reuse these seeds to generate the same Gaussian samples. *seeds*[$i$] stores the seed used for generating $i$-th sample in the RS execution of $f$, and *predictions*[$i$] stores the prediction of $f$ on the corrsponding $x + \epsilon$. We evaluate $f^p$ on each corruption $\epsilon$ generated from *seeds* and match them to predictions by $f$. $c_f$ and $c_{f^p}$ represent the top class prediction by $f$ and $f^p$ respectively. $n_\Delta$ is the count of the number of corruptions $\epsilon$ such that $f$ and $f^p$ do not match on $x + \epsilon$.

---

**Algorithm 3** Estimate $\zeta_x$

**Inputs:** $f^p$: DNN obtained from approximating $f$, $\sigma$: standard deviation, $x$: input to the DNN, $n_p$: number of Gaussian samples used for estimating $\zeta_x$, $\mathcal{C}_f$: stores the information to be reused from certification of $f$, $\alpha_\zeta$: confidence parameter
**Output:** Estimated value of $\zeta_x$

1: **function** ESTIMATEZETA($f^p, \sigma, x, n_p, \mathcal{C}_f, \alpha_\zeta$)
2:    $n_\Delta \leftarrow 0$
3:    $seeds \leftarrow$ seeds for original samples from $\mathcal{C}_f[x]$
4:    $predictions \leftarrow f$'s predictions on samples from $\mathcal{C}_f[x]$
5:    **for** $i \in \{1, \ldots n_p\}$ **do**
6:      $\epsilon \sim \mathcal{N}(0, \sigma^2 I)$ using $seeds[i]$
7:      $c_f \leftarrow predictions[i]$
8:      $c_{f^p} \leftarrow f^p(x + \epsilon)$
9:      $n_\Delta \leftarrow n_\Delta + I(c_f \neq c_{f^p})$
10:   **return** UpperConfidenceBound($n_\Delta, n_p, 1 - \alpha_\zeta$)

---

The function UpperConfidenceBound($n_\Delta, n_p, 1 - \alpha_\zeta$) in the pseudocode returns a one-sided $1 - \alpha_\zeta$ upper confidence interval for the Binomial parameter $p$ given a sample $n_\Delta \sim Binomial(n_p, p)$. We compute this upper confidence bound using the Clopper-Pearson method. This is similar to how the lower confidence bound is computed in the standard RS Algorithm 1. It is sound since the Clopper-Pearson method is conservative.

Reusing the seeds for generating noisy samples does not change the certified radius and is 2x faster compared to naive Monte Carlo estimation of $\zeta_x$ with fresh Gaussian samples. Storing the seeds used in the cache results in a small memory overhead (less than 2MBs for our largest benchmark). We use the same Gaussian samples for estimations of $\underline{p_A}$ and $\zeta_x$. This is equivalent to estimating two functions, $p(X)$ and $q(X)$, of a random variable $X$, where the same set of samples of $X$ can be employed for their respective estimations. Theorem 4 makes no assumptions about the independence of estimating $\underline{p_A}$ and $\zeta_x$, thus we can soundly reuse the same Gaussian samples for both estimations.

## 4 EXPERIMENTAL METHODOLOGY

**Networks and Datasets.** We evaluate IRS on CIFAR-10 (Krizhevsky et al.) and ImageNet (Deng et al., 2009). On each dataset, we use several classifiers, each with a different $\sigma$'s. For an experiment that adds Gaussian corruptions with $\sigma$ to the input, we use the network that is trained with Gaussian augmentation with variance $\sigma^2$. On CIFAR-10 we use the base classifier a 20-layer and 110-layer residual network. On ImageNet our base classifier is a ResNet-50.

**Network Approximations.** We evaluate IRS on multiple approximations. We consider float16 (fp16), bfloat16 (bf16), and int8 quantizations (Section 5.1). We show the effectiveness of IRS on pruning approximation in Section 5.3. For int8 quantization, we use dynamic per-channel quantization mode. from (Paszke et al., 2019) library. For float16 and bfloat16 quantization, we change the data type of the DNN weights from float32 to the respective types. We perform float32, float16, and bfloat16 inferences on the GPU and int8 inferences on CPU since Pytorch does not support int8 quantization for GPUs yet (PyTorch). For the pruning experiment, we perform the lowest weight magnitude (LWM) pruning. The top-1 accuracy of the networks used in the evaluation and the approximate networks is discussed in Appendix A.3.

**Experimental Setup.** We ran experiments on a 48-core Intel Xeon Silver 4214R CPU with 2 NVidia RTX A5000 GPUs. IRS is implemented in Python and uses PyTorch 2.0.1. (Paszke et al., 2019).

**Hyperparameters.** We use confidence parameters $\alpha = 0.001$ for the certification of $g$, and $\alpha_\zeta = 0.001$ for the estimation of $\zeta_x$. To establish a fair comparison, we set the baseline confidence with $\alpha_b = \alpha + \alpha_\zeta = 0.002$. This choice ensures that both the baseline and IRS, provide certified radii with equal confidence. We use grid search to choose an effective value for $\gamma$. A detailed description of our hyperparameter search and its results are described in Section 5.4.

**Average Certified Radius.** We compute the certified radius $r$ when the certification algorithm did not abstain and returned the correct class with radius $r$, for both IRS (Algorithm 2) and the baseline (Algorithm 1). In other cases, we say that the certified radius $r = 0$. We compute the *average certified radius* (ACR) by taking the mean of certified radii computed for inputs in the test set. Higher ACR indicates stronger robustness certification guarantees.

**Speedup.** IRS is applicable while certifying multiple similar networks, where it can reuse the certification of one of the networks for faster certification of all other similar networks. We demonstrate the effectiveness of IRS by comparing IRS's certification time for these other similar networks with the baseline certification from scratch. We do not include the certification time of the first network in the comparison as it adds the same time for both IRS and baseline.

## 5 EXPERIMENTAL RESULTS

We now present our main evaluation results. We consider the float32 representation of the DNN as $f$ and a particular approximation as $f^p$. However, IRS can be used with any similar $f$ and $f^p$s, e.g., where $f$ is an int8 quantized network and $f^p$ is the float32 network. In all of our experiments, we follow a specific procedure:

1. We certify the smoothed classifier $g$ using standard RS with a sample size of $n$.
2. We approximate the base classifier $f$ with $f^p$.
3. Using the IRS, we certify smoothed classifier $g^p$ by employing Algorithm 2 and utilizing the cached information $\mathcal{C}_f$ obtained from the certification of $g$.

We compare IRS to the baseline that uses standard non-incremental RS (Algorithm 1), to certify $g^p$. Our results compare ACR and certification time between IRS and the baseline for various $n_p$ values.

## 5.1 EFFECTIVENESS OF IRS

We compare the ACR and the certification time of the baseline and IRS for the common int8 quantization. We use $n = 10^5$ samples for certification of $g$. For certifying $g^p$, we consider $n_p$ values from $\{5\%, \dots 50\%\}$ of $n$ and $\sigma = 1$. We perform experiments on $500$ images and compute the total time for certifying $g^p$.

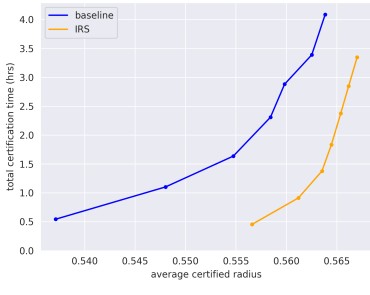

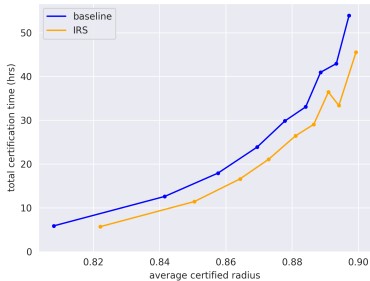

(a) ResNet-110 on CIFAR-10

(b) ResNet-50 on ImageNet

Figure 3: Total certification time versus ACR with $\sigma = 1.0$.

Figure 3 presents the comparison between IRS and RS for int8 quantization. The x-axis displays the ACR and the y-axis displays the certification time. The plot consists of 10 markers each for the IRS and the baseline representing a specific value of $n_p$. Expectedly, the higher the value of $n_p$, the higher the average time and ACR. The marker coordinate denotes the ACR and the time for an experiment. In all the cases, IRS consistently takes less certification time to obtain the same ACR.

Figure 3a, for ResNet-110 on CIFAR10, shows that IRS reduced the certification time from 2.91 hours (baseline) to 0.71 hours, resulting in time savings of 2.12 hours (4.1x faster). Moreover, we see that IRS achieves an ACR of more than 0.565, whereas the baseline does not reach this ACR for any of the $n_p$ values in our experiments.

Figure 3b, for ResNet-50 on ImageNet, for certifying an ACR of 0.875, IRS substantially reduced certification time from 27.82 hours (baseline) to 22.45 hours, saving approximately 5.36 hours (1.24x faster). Additionally, IRS achieved an ACR of 0.90 and reduced the certification time from 53.93 hours (baseline) to 40.58 hours, resulting in substantial time savings of 13.35 hours (1.33x faster).

## 5.2 IRS SPEEDUPS ON DIFFERENT QUANTIZATIONS

Next, we study if IRS can handle other kinds of quantization. We perform experiments for 10 different values of $n_p$ along with distinct approximations, and 3 values of $\sigma$. Since this would take months of experiment time with $n$ and $n_p$ values from Section 5.1, for the rest of the experiments we use smaller values for these parameters. In these experiments, we compute the relative speedup due to IRS in comparison to the baseline. We use $n = 10^4$ for samples for certification of $g$. For certifying $g^p$, we consider $n_p$ values from $\{1\%, \dots 10\%\}$ of $n$. For CIFAR10, we consider $\sigma \in \{0.25, 0.5, 1.0\}$, and for Ima-

Table 2: Average IRS speedup for combinations of quantizations and $\sigma$'s.

| Dataset | Architecture | $\sigma$ | Quantization | | |
|---|---|---|---|---|---|
| | | | fp16 | bf16 | int8 |
| CIFAR10 | ResNet-20 | 0.25 | 1.37x | 1.29x | 1.3x |
| | | 0.5 | 1.79x | 1.7x | 1.77x |
| | | 1.0 | 2.85x | 2.41x | 2.65x |
| CIFAR10 | ResNet-110 | 0.25 | 1.42x | 1.35x | 1.29x |
| | | 0.5 | 1.97x | 1.74x | 1.77x |
| | | 1.0 | 3.02x | 2.6x | 2.6x |
| ImageNet | ResNet-50 | 0.5 | 1.2x | 1.14x | 1.19x |
| | | 1.0 | 1.43x | 1.31x | 1.43x |
| | | 2.0 | 2.04x | 1.69x | 1.80x |

geNet, we consider $\sigma \in \{0.5, 1.0, 2.0\}$ as in the previous work(Cohen et al., 2019). We validated that the speedups for int8 quantization in this section for ResNet-50-ImageNet and ResNet-110-CIFAR10 are similar to those studied in Section 5.1.

To quantify IRS's average speedup over the baseline, we employ an approximate area under the curve (AOC) analysis. Specifically, we plot the certification time against the ACR. In most cases, IRS

certifies a larger ACR compared to the baseline, resulting in regions on the x-axis where IRS exists but the baseline does not. To ensure a conservative estimation, we calculate the speedup only within the range where both IRS and the baseline exist. We determine the speedup by computing the ratio of the AOC for IRS to the AOC for the baseline within this common range. Table 2 summarizes the average speedups for all quantization experiments.

We observe that IRS gets a larger speedup for smoothing with larger $\sigma$ since on average the $p_A$ values are smaller. Appendix A.7 presents a further justification for this observation. Appendix A.9 presents further experiments with all combinations of DNNs, $\sigma$, and quantizations.

## 5.3 IRS SPEEDUPS ON PRUNED MODELS

In this experiment, we study IRS's ability to certify beyond quantized models. We employ $l_1$ unstructured pruning, which prunes the fraction of the lowest $l_1$ magnitude weights from the DNN. Table 3 presents the average IRS speedup for DNNs obtained by pruning $5\%, 10\%$ and $20\%$ weights. The speedups range from 0.99x to 2.7x. As the DNN is pruned more aggressively, it's expected that IRS's speedup will be lower. This is due to higher values of $\zeta_x$ associated with aggressive pruning. In Appendix A.4, we provide average $\zeta_x$ values for all approximations. Compared to pruning, quantization typically yields smaller $\zeta_x$ values, making IRS more effective for quantization.

Table 3: Average IRS speedup for combinations of pruning ratio and $\sigma$'s.

| Dataset | Architecture | $\sigma$ | Prune | | |
| --- | --- | --- | --- | --- | --- |
| | | | 5% | 10% | 20% |
| CIFAR10 | ResNet-20 | 0.25 | 1.3x | 1.25x | 0.99x |
| | | 0.5 | 1.63x | 1.39x | 1.13x |
| | | 1.0 | 2.5x | 2.09x | 1.39x |
| CIFAR10 | ResNet-110 | 0.25 | 1.35x | 1.24x | 1.04x |
| | | 0.5 | 1.83x | 1.6x | 1.23x |
| | | 1.0 | 2.7x | 2.25x | 1.63x |
| ImageNet | ResNet-50 | 0.5 | 1.19x | 1.04x | 0.87x |
| | | 1.0 | 1.36x | 1.15x | 0.87x |
| | | 2.0 | 1.87x | 1.54x | 1.01x |

## 5.4 ABLATION STUDIES

Next, we show the effect of $\gamma$ on ACR. In Appendix A.5 we show IRS speedup on distinct values of $n$.

**Sensitivity to threshold $\gamma$.** For each DNN architecture, we chose the hyperparameter $\gamma$ by running IRS to certify a small subset of the validation set images for certifying the int8 quantized DNN and comparing the ACR. The choice of $\gamma$ has no effect on certification time, as we perform $n_p$ inferences in both cases, $p_A < \gamma$ and $p_A > \gamma$. We use the same $\gamma$ for each DNN irrespective of the approximation and $\sigma$. We use the grid search to choose the best value of gamma from the set

Table 4: ACR for each $\gamma$.

| $\gamma$ | CIFAR10 ResNet-20 | CIFAR10 ResNet-110 | ImageNet ResNet-50 |
| --- | --- | --- | --- |
| 0.9 | 0.438 | 0.436 | 0.458 |
| 0.95 | 0.442 | 0.439 | 0.464 |
| 0.975 | 0.445 | 0.441 | 0.465 |
| 0.99 | **0.446** | **0.443** | 0.466 |
| 0.995 | 0.445 | 0.442 | **0.467** |
| 0.999 | 0.444 | 0.442 | 0.464 |

$\{0.9, 0.95, 0.975, 0.99, 0.999\}$. Table 4 presents the ACR obtained for each $\gamma$. We chose $\gamma$ as 0.99 for CIFAR10 networks and 0.995 for the ImageNet networks since they result in the highest ACR.

## 6 RELATED WORK

**Incremental Program Verification.** The scalability of traditional program verification has been significantly improved by incremental verification, which has been applied on an industrial scale (Johnson et al., 2013; O'Hearn, 2018; Stein et al., 2021). Incremental program analysis tasks achieve faster analysis of individual commits by reusing partial results (Yang et al., 2009), constraints (Visser et al., 2012), and precision information (Beyer et al., 2013) from previous runs.

**Incremental DNN Certification.** Several methods have been introduced in recent years to certify the properties of DNNs deterministically (Tjeng et al., 2017; Bunel et al., 2020; Katz et al., 2017; Wang et al., 2021b; Laurel et al., 2022; 2023) and probabilisticly (Cohen et al., 2019). Researchers used incremental certification speed up DNN certification (Fischer et al., 2022b; Ugare et al., 2022; Wei and Liu, 2021; Ugare et al., 2023; Ugare et al.) – these works apply complete and incomplete

deterministic certification using formal logic cannot scale to e.g., ImageNet. In contrast, we propose incremental probabilistic certification with Randomized Smoothing, which enables much greater scalability.

**Randomized Smoothing.** Cohen et al. (2019) introduced the addition of Gaussian noise to achieve $l_2$-robustness results. Several extensions to this technique utilize different types of noise distributions and radius calculations to determine certificates for general $l_p$-balls. Yang et al. (2020) and Zhang et al. (2020) derived recipes for determining certificates for $p = 1, 2$, and $\infty$. Lee et al. (2019), Wang et al. (2021a), and Schuchardt et al. (2021) presented extensions to discrete perturbations such as $l_0$-perturbations, while Bojchevski et al. (2023), Gao et al. (2020), Levine and Feizi (2020), and Liu et al. (2021) explored extensions to graphs, patches, and point cloud manipulations. Dvijotham et al. (2020) presented theoretical derivations for the application of both continuous and discrete smoothing measures, while Mohapatra et al. (2020) improved certificates by using gradient information. Horváth et al. (2022) used ensembles to improve the certificate.

Beyond norm-balls certificates, Fischer et al. (2020) and Li et al. (2021) presented how geometric operations such as rotation or translation can be certified via Randomized Smoothing. yeh Chiang et al. (2022) and Fischer et al. (2022a) demonstrated how the certificates can be extended from the setting of classification to regression (and object detection) and segmentation, respectively. For classification, Jia et al. (2020) extended certificates from just the top-1 class to the top-k classes, while Kumar et al. (2020) certified the confidence of the classifier, not just the top-class prediction. Rosenfeld et al. (2020) used Randomized Smoothing to defend against data poisoning attacks. These RS extensions (using different noise distributions, perturbations, and geometric operations) are orthogonal to the standard RS approach from Cohen et al. (2019). While these extensions have been shown to improve the overall bredth of RS, IRS is complementary to these extensions.

## 7 LIMITATIONS

We showed that IRS is effective at certifying the smoothed version of the approximated DNN. However, there are certain limitations to the effectiveness of IRS. First, the IRS algorithm requires a cache with the top predicted class index, its lower confidence bound, and the seeds for Gaussian corruptions obtained from the RS execution of the original classifier. However, storing this additional information is reasonable since it has negligible memory overhead and is a byproduct of certification (as trustworthy ML matures, we anticipate that this information will be shipped with pre-certified networks for reproducibility purposes).

The smoothing parameter $\sigma$ used in IRS affects its efficiency, with larger values of $\sigma$ generally leading to better results. As a consequence, we observed a smaller speedup when using a smaller value of $\sigma$ (e.g., 0.25 on CIFAR10) compared to a larger value (e.g., 1 on CIFAR10). The value of $\sigma$ offers a trade-off between robustness and accuracy. By choosing a larger $\sigma$, one can improve robustness but it may lead to a loss of accuracy in the model.

IRS targets fast certification while maintaining a sufficiently large radius. Therefore, we considered $n_p$ smaller than $50\%$ of $n$ for our evaluation. However, IRS certified radius can be smaller than the non-incremental RS, provided the user has a larger sample budget. In our experiment in Appendix A.6 we test IRS on larger $n_p$ and observe that IRS is better than baseline for $n_p$ less than $70\%$ of $n$. This is particularly advantageous when computational resources are limited.

## 8 CONCLUSION

We propose IRS, the first incremental approach for probabilistic DNN certification. IRS leverages the certification guarantees obtained from the smoothed model to certify a smoothed approximated model with very few samples. Reusing the computation of original guarantees significantly reduces the computational cost of certification while maintaining strong robustness guarantees. IRS speeds up certification up to 4.1x over the standard non-incremental RS baseline on state-of-the-art classification models. We anticipate that IRS can be particularly useful for approximate tuning when the users need to analyze the robustness of multiple similar networks. Further, one can easily ship the certification cache to allow other users to further modify these networks based on their specific device and application needs and recertify the new network. We believe that our approach paves the way for efficient and effective certification of DNNs in real-world applications.

## ACKNOWLEDGMENTS

We thank the anonymous reviewers for their comments. This research was supported in part by NSF Grants No. CCF-1846354, CCF-2217144, CCF-2238079, CCF-2313028, CCF-2316233, CNS-2148583, USDA NIFA Grant No. NIFA-2024827 and Google Research Scholar award.

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

## A    APPENDIX

### A.1    OBSERVATION FOR BINOMIAL CONFIDENCE INTERVAL METHODS

In this section, we show the plots for the number of samples required to estimate an unknown binomial proportion parameter through two popular estimation techniques - the Wilson (Wilson, 1927) and Agresti-Coull method (Agresti and Coull, 1998). For this experiment, we use three different values of the target error $\chi = 0.5$ %, 0.75 %, and 1.0 % and a fixed confidence value $(1 - \alpha) = 0.99$ for both estimation methods. As shown in Fig 4, for a fixed target error $\chi$, confidence $(1 - \alpha)$, and estimation technique, the number of samples required for estimation peaks, when the actual parameter value is around $0.5$ and is the smallest around the boundaries. This is consistent with the observation described in Section 3.1.

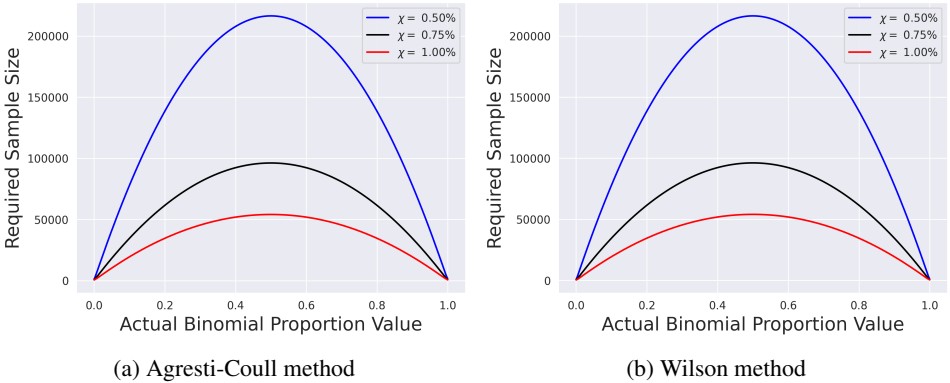

(a) Agresti-Coull method                    (b) Wilson method

Figure 4: The number of samples for the Agresti-Coull and Wilson method to achieve a target error $\chi$ with confidence $(1 - \alpha)$ where $\alpha = 0.01$. The plots show that the number of required samples for different methods peaks at 0.5 and decreases towards the boundaries.

### A.2    THEOREMS

**Theorem 2.** *If a classifier $f^p$ is such that for all $x \in \mathbb{R}^m$, $\mathbb{P}_\epsilon(f(x + \epsilon) \neq f^p(x + \epsilon)) \leq \zeta_x$, and classifier $f$ satisfies $\mathbb{P}_\epsilon(f(x + \epsilon) = c_A) \geq \underline{p_A} \geq \overline{p_B} \geq \max_{c \neq c_A} \mathbb{P}_\epsilon(f(x + \epsilon) = c)$ and $\underline{p_A} - \zeta_x \geq \overline{p_B} + \zeta_x$ then $g^p$ satisfies $g^p(x + \delta) = c_A$ for all $\delta$ satisying $\|\delta\|_2 \leq \frac{\sigma}{2}(\Phi^{-1}(\underline{p_A} - \zeta_x) - \Phi^{-1}(\overline{p_B} + \zeta_x))$*

*Proof.* If $f(x + \epsilon) = c_A$ and $f^p(x + \epsilon) = f(x + \epsilon)$ then $f^p(x + \epsilon) = c_A$.
Thus, if $f^p(x + \epsilon) \neq c_A$ then $f(x + \epsilon) \neq c_A$ or $f^p(x + \epsilon) \neq f(x + \epsilon)$.
Using union bound,

$$\mathbb{P}_\epsilon(f^p(x + \epsilon) \neq c_A) \leq \mathbb{P}_\epsilon(f(x + \epsilon) \neq c_A) + \mathbb{P}_\epsilon(f(x + \epsilon) \neq f^p(x + \epsilon))$$
$$(1 - \mathbb{P}_\epsilon(f^p(x + \epsilon) = c_A)) \leq (1 - \mathbb{P}_\epsilon(f(x + \epsilon) = c_A)) + \mathbb{P}_\epsilon(f(x + \epsilon) \neq f^p(x + \epsilon))$$
$$\mathbb{P}_\epsilon(f(x + \epsilon) = c_A) \leq \mathbb{P}_\epsilon(f^p(x + \epsilon) = c_A) + \mathbb{P}_\epsilon(f(x + \epsilon) \neq f^p(x + \epsilon))$$
$$\underline{p_A} - \zeta_x \leq \mathbb{P}_\epsilon(f^p(x + \epsilon) = c_A)$$

Similarly, if $f(x + \epsilon) \neq c$ then $f^p(x + \epsilon) \neq c$ or $f^p(x + \epsilon) \neq f(x + \epsilon)$.
Hence, using union bound,

$$\mathbb{P}_\epsilon(f(x + \epsilon) \neq c) \leq \mathbb{P}_\epsilon(f^p(x + \epsilon) \neq c) + \mathbb{P}_\epsilon(f(x + \epsilon) \neq f^p(x + \epsilon))$$
$$(1 - \mathbb{P}_\epsilon(f(x + \epsilon) = c)) \leq (1 - \mathbb{P}_\epsilon(f^p(x + \epsilon) = c)) + \mathbb{P}_\epsilon(f(x + \epsilon) \neq f^p(x + \epsilon))$$
$$\mathbb{P}_\epsilon(f^p(x + \epsilon) = c) \leq \mathbb{P}_\epsilon(f(x + \epsilon) = c) + \mathbb{P}_\epsilon(f(x + \epsilon) \neq f^p(x + \epsilon))$$
$$\max_{c \neq c_A} \mathbb{P}_\epsilon(f^p(x + \epsilon) = c) \leq \max_{c \neq c_A} \mathbb{P}_\epsilon(f(x + \epsilon) = c) + \zeta_x$$
$$\max_{c \neq c_A} \mathbb{P}_\epsilon(f^p(x + \epsilon) = c) \leq \overline{p_B} + \zeta_x$$

Hence, using Theorem 1, $g^p$ satisfies $g^p(x + \delta) = c_A$ for all $\delta$ satisying $\|\delta\|_2 \leq \frac{\sigma}{2}(\Phi^{-1}(\underline{p_A} - \zeta_x) - \Phi^{-1}(\overline{p_B} + \zeta_x))$

□

**Theorem 3.** *If $\underline{p_A} - \zeta_x \geq \frac{1}{2}$, then $\sigma\Phi^{-1}(\underline{p_A} - \zeta_x) \leq \frac{\sigma}{2}(\Phi^{-1}(\underline{p_A} - \zeta_x) - \Phi^{-1}(\overline{p_B} + \zeta_x))$*

*Proof.* Since $\underline{p_A} - \zeta_x \geq \frac{1}{2}$, $0 \leq \underline{p_A} \leq 1$ and $\zeta_x \geq 0$, we get $0 \leq \underline{p_A} - \zeta_x \leq 1$

And since $1 - \underline{p_A} \geq \overline{p_B}$, we get $\overline{p_B} + \zeta_x \leq \frac{1}{2}$, and thus, $0 \leq \overline{p_B} + \zeta_x \leq 1$

Since $\Phi^{-1}(1 - t) = -\Phi^{-1}(t)$

$$\Phi^{-1}(\overline{p_B} + \zeta_x) = -\Phi^{-1}(1 - (\overline{p_B} + \zeta_x))$$

$$= -\Phi^{-1}((1 - \overline{p_B}) - \zeta_x)$$

Since $1 - \underline{p_A} \geq \overline{p_B}$

$$\leq -\Phi^{-1}(\underline{p_A} - \zeta_x)$$

Hence,

$$\Phi^{-1}(\underline{p_A} - \zeta_x) \leq -\Phi^{-1}(\overline{p_B} + \zeta_x)$$

$$\frac{\sigma}{2}\Phi^{-1}(\underline{p_A} - \zeta_x) \leq -\frac{\sigma}{2}\Phi^{-1}(\overline{p_B} + \zeta_x)$$

Adding $\frac{\sigma}{2}\Phi^{-1}(\underline{p_A} - \zeta_x)$ on both sides,

$$\sigma\Phi^{-1}(\underline{p_A} - \zeta_x) \leq \frac{\sigma}{2}(\Phi^{-1}(\underline{p_A} - \zeta_x) - \Phi^{-1}(\overline{p_B} + \zeta_x))$$

$\square$

**Theorem 4.** *If $\mathbb{P}_\epsilon(f(x + \epsilon) = f^p(x + \epsilon)) > 1 - \zeta_x$ with confidence at least $1 - \alpha_\zeta$. If classifier $f$ satisfies $\mathbb{P}_\epsilon(f(x + \epsilon) = c_A) \geq \underline{p_A}$ with confidence at least $1 - \alpha$. Then for classifier $f^p$, $\mathbb{P}_\epsilon(f^p(x + \epsilon) = c_A) \geq \underline{p_A} - \zeta_x$ with confidence at least $1 - (\alpha + \alpha_\zeta)$*

*Proof.* Suppose $f$ and $f^p$ are classifiers such that for a fixed $x \in \mathbb{R}^m$, $\mathbb{P}_\epsilon(f(x + \epsilon) = c_A) \geq \underline{p_A}$ and $\mathbb{P}_\epsilon(f(x + \epsilon) = f^p(x + \epsilon)) > 1 - \zeta_x$. Note that this is true by the definition of $\underline{p_A}$, and is a separate $\underline{p_A}$ for each $x$. The statement is not true for all $x$ with single $\underline{p_A}$

Let $E_1$ denote the event that $\mathbb{P}_\epsilon(f(x + \epsilon) = c_A) \geq \underline{p_A}$.

Let $E_2$ denote the event that $\mathbb{P}_\epsilon(f(x + \epsilon) = f^p(x + \epsilon)) > 1 - \zeta_x$.

By Theorem 2,

$$\mathbb{P}_\epsilon(f(x + \epsilon) = c_A) \leq \mathbb{P}_\epsilon(f^p(x + \epsilon) = c_A) + \mathbb{P}_\epsilon(f(x + \epsilon) \neq f^p(x + \epsilon))$$

$$\underline{p_A} - \zeta_x \leq \mathbb{P}_\epsilon(f^p(x + \epsilon) = c_A)$$

Let $E_3$ denote the event that $\underline{p_A} - \zeta_x \leq \mathbb{P}_\epsilon(f^p(x + \epsilon) = c_A)$

Since, $E_1$ and $E_2$ imply $E_3$ i.e. $E_1 \cap E_2 \subseteq E_3$,

$$\mathbb{P}(E_3) \geq \mathbb{P}(E_1 \cap E_2)$$

By the additive rule of probability,

$$\mathbb{P}(E_1 \cap E_2) = \mathbb{P}(E_1) + \mathbb{P}(E_2) - \mathbb{P}(E_1 \cup E_2)$$

$$\mathbb{P}(E_3) \geq (1 - \alpha) + (1 - \alpha_\zeta) - 1$$

$$\mathbb{P}(E_3) \geq 1 - (\alpha + \alpha_\zeta)$$

Hence, for classifier $f^p$, $\mathbb{P}_\epsilon(f^p(x + \epsilon) = c_A) \geq \underline{p_A} - \zeta_x$ has confidence at least $1 - (\alpha + \alpha_\zeta)$ $\square$

## A.3 Evaluation Networks

Table 5 and Table 6 respectively present the standard top-1 accuracy of the original and approximated base classifiers and smoothed classifiers respectively.

Table 5: Standard top-1 accuracy for (non-smoothed) networks for combinations of approximations and $\sigma$'s.

| Dataset | Architecture | $\sigma$ | original | Quantization | | | Prune | | |
|---|---|---|---|---|---|---|---|---|---|
| | | | | fp16 | bf16 | int8 | 5% | 10% | 20% |
| | | 0.25 | 67.2 | 67.2 | 66.8 | 67.2 | 67.4 | 66.6 | 66.6 |
| CIFAR10 | ResNet-20 | 0.5 | 56.8 | 56.8 | 57.2 | 56.8 | 57 | 57.4 | 58 |
| | | 1.0 | 47.2 | 47.2 | 47.0 | 47.2 | 47 | 46.2 | 45.2 |
| | | 0.25 | 69.0 | 69.0 | 69.4 | 69.0 | 69.2 | 68.8 | 68.2 |
| CIFAR10 | ResNet-110 | 0.5 | 59.4 | 59.4 | 59.4 | 59.4 | 59.6 | 59 | 58.8 |
| | | 1.0 | 47.0 | 47.0 | 46.8 | 46.8 | 46.8 | 47.2 | 47 |
| | | 0.5 | 24.2 | 24.2 | 24.4 | 24.2 | 24.2 | 24.4 | 24.2 |
| ImageNet | ResNet-50 | 1.0 | 9.6 | 9.6 | 9.6 | 9.6 | 9.6 | 9.6 | 9.6 |
| | | 2.0 | 6.4 | 6.4 | 6.4 | 6.4 | 6.4 | 6.4 | 6.4 |

Table 6: standard top-1 accuracy for smoothed networks for combinations of approximations and $\sigma$'s.

| Dataset | Architecture | $\sigma$ | original | Quantization | | | Prune | | |
|---|---|---|---|---|---|---|---|---|---|
| | | | | fp16 | bf16 | int8 | 5% | 10% | 20% |
| | | 0.25 | 77.2 | 77 | 77.2 | 77.2 | 77.6 | 77.2 | 77.6 |
| CIFAR10 | ResNet-20 | 0.5 | 67.8 | 67.4 | 67.8 | 67.8 | 67.8 | 67.4 | 67.8 |
| | | 1.0 | 55.6 | 55.6 | 55.6 | 55.8 | 54.8 | 55.2 | 55.6 |
| | | 0.25 | 76.6 | 76.4 | 76.2 | 76.4 | 76.2 | 76.2 | 76.4 |
| CIFAR10 | ResNet-110 | 0.5 | 66.2 | 67 | 68 | 66.4 | 67 | 66.8 | 66.6 |
| | | 1.0 | 55.6 | 55.4 | 56.2 | 56.2 | 55 | 55 | 54.8 |
| | | 0.5 | 63.8 | 63.4 | 63.2 | 63.4 | 63.6 | 64 | 63 |
| ImageNet | ResNet-50 | 1.0 | 48.8 | 48.6 | 48.8 | 48.6 | 48.8 | 48.6 | 47.8 |
| | | 2.0 | 34.4 | 34.2 | 33.8 | 34.2 | 34.2 | 34.4 | 33.4 |

Table 8: $\zeta_x$ for approximate networks trained on different Gaussian augmentation $\sigma$'s.

| Dataset | Architecture | $\sigma$ | Quantization | | | Prune | | |
|---|---|---|---|---|---|---|---|---|
| | | | fp16 | bf16 | int8 | 5% | 10% | 20% |
| CIFAR10 | ResNet-20 | 0.25 | 0.01 | 0.01 | 0.006 | 0.01 | 0.02 | 0.04 |
| | | 0.5 | 0.006 | 0.008 | 0.01 | 0.01 | 0.02 | 0.03 |
| | | 1.0 | 0.006 | 0.007 | 0.006 | 0.007 | 0.02 | 0.02 |
| CIFAR10 | ResNet-110 | 0.25 | 0.006 | 0.01 | 0.006 | 0.009 | 0.02 | 0.04 |
| | | 0.5 | 0.006 | 0.006 | 0.006 | 0.008 | 0.02 | 0.03 |
| | | 1.0 | 0.006 | 0.008 | 0.009 | 0.007 | 0.01 | 0.02 |
| ImageNet | ResNet-50 | 0.5 | 0.006 | 0.009 | 0.006 | 0.01 | 0.02 | 0.09 |
| | | 1.0 | 0.007 | 0.01 | 0.006 | 0.01 | 0.02 | 0.08 |
| | | 2.0 | 0.006 | 0.01 | 0.006 | 0.007 | 0.02 | 0.07 |

## A.4 $\zeta_x$ EVALUATION

We compute $\zeta_x$ value as the binomial confidence upper limit using (Clopper and Pearson, 1934) method with $n = 1000$ samples. For an experiment that adds Gaussian corruptions with $\sigma$ to the input, we use the network that is trained with Gaussian data augmentation with variance $\sigma^2$.

## A.5 SENSITIVITY TO CHANGING $n$

In Section 5, to save time due to a large number of approximations and DNNs tested, we used $n = 10^4$ samples for $g$'s certification. Here, we present the effect of certifying with a larger $n$ by comparing the ACR vs certification time on the IRS and baseline approaches for ResNet-20 on CIFAR10. On average, for larger $n$, we demonstrate greater speedup for larger $\sigma$. For instance, for int8 quantization with $\sigma = 1.0$, the speedup for certifying with $n = 10^6$ samples was 5.85x as compared to certification with $n = 10^4$ which yielded at 2.65x speedup. However, for smaller $\sigma$, certification with a larger n results in less speedup. For $\sigma = 0.25$, we observe speedups from 1.29x to 1.37x for $n = 10^4$ whereas from 0.93x to 1.15x for $n = 10^6$.

Table 7: Average IRS speedup for combinations of $n$, $\sigma$'s, and quantizations for ResNet-20 on CIFAR10.

| $n$ | $\sigma$ | Quantization | | |
|---|---|---|---|---|
| | | fp16 | bf16 | int8 |
| $10^4$ | 0.25 | 1.37x | 1.29x | 1.3x |
| | 0.5 | 1.79x | 1.7x | 1.77x |
| | 1.0 | 2.85x | 2.41x | 2.65x |
| $10^5$ | 0.25 | 1.22x | 1.11x | 1.27x |
| | 0.5 | 1.73x | 1.4x | 1.86x |
| | 1.0 | 3.88x | 2.40x | 4.31x |
| $10^6$ | 0.25 | 1.12x | 0.93x | 1.15x |
| | 0.5 | 1.97x | 1.04x | 2.25x |
| | 1.0 | 4.58x | 1.25x | 5.85x |

## A.6 EVALUATION WITH LARGER $n_p$

The objective of IRS is to certify the approximated DNN with few samples. Thus, we consider $n_p$ ranging from 1% to 10%. Nevertheless, we check IRS effectiveness for larger $n_p$ values in this ablation study.

Since, IRS certifies radius $\sigma \Phi^{-1}(\underline{p_A} - \zeta_x)$ that is always smaller than original certified radius. When $n_p = n$, the baseline running from scratch should perform better than IRS, as it will reach a certification radius close to $\sigma \Phi^{-1}(\underline{p_A})$.

In this experiment, on CIFAR10 ResNet-20 with $\sigma = 1$, we let $n_p \in \{5\%, 10\% \ldots 80\%\}$ of $n$. Figure 5 shows the ACR vs mean time plot for the baseline and IRS. We see that IRS gives speedup for $n_p = 70\%$. For $n_p = 75\%$ and $n_p = 80\%$, we see that baseline ACR is higher and IRS cannot achieve that ACR.

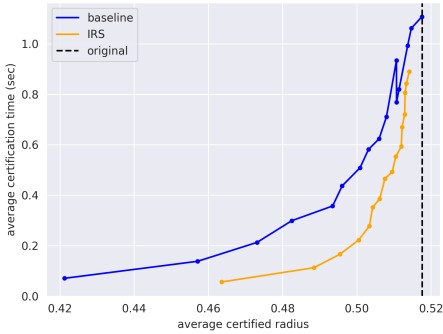

Figure 5: CIFAR10 ResNet-20 with $\sigma = 1$, for $n_p \in \{5\%, 10\% \ldots 80\%\}$ of $n$

## A.7    Effect of standard deviation $\sigma$ on IRS speedup.

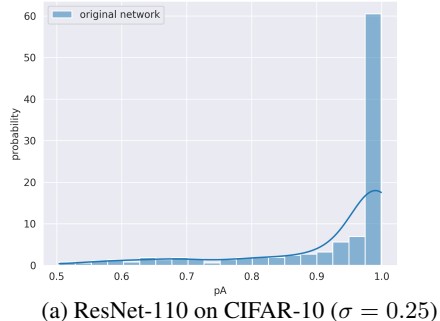
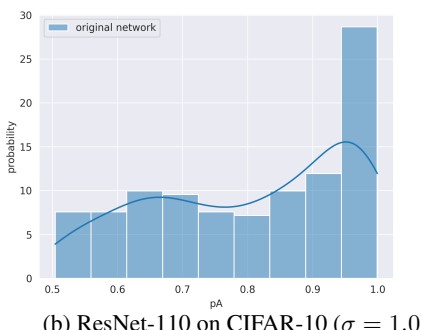

(a) ResNet-110 on CIFAR-10 ($\sigma = 0.25$)        (b) ResNet-110 on CIFAR-10 ($\sigma = 1.0$)

Figure 6: Distribution of $\underline{p_A}$ values greater than 0.5 with different $\sigma$ for ResNet-110 on CIFAR-10.

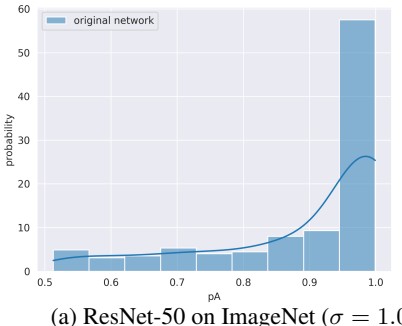
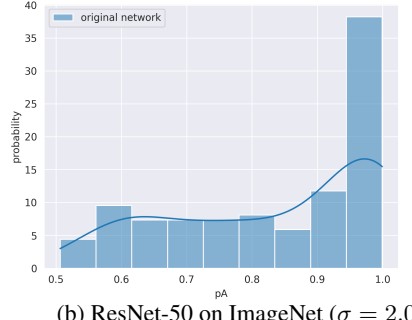

(a) ResNet-50 on ImageNet ($\sigma = 1.0$)        (b) ResNet-50 on ImageNet ($\sigma = 2.0$)

Figure 7: Distribution of $\underline{p_A}$ values greater than 0.5 with different $\sigma$ for ResNet-50 on ImageNet.

Figure 6 and Figure 7, present the $\underline{p_A}$ distribution between $0.5$ to $1$, for ResNet-110 on CIFAR-10 and ResNet-50 on ImageNet respectively. The x-axis represents the range of $\underline{p_A}$ values and the y-axis represents their respective proportion. The results show that while certifying larger $\sigma$, on average the $\underline{p_A}$ values are smaller. As shown in Figure 7a, for $\sigma = 0.25$, less than $35\%$ of $\underline{p_A}$ values are smaller than $0.95$. On the other hand, in Figure 7b, when $\sigma = 1.0$, the distribution is less left-skewed as nearly $75\%$ of $\underline{p_A}$ values are less than $0.95$. When the $\sigma$ is larger, the values of $\underline{p_A}$ tend to be farther away from 1. Therefore, the estimation of $\underline{p_A}$ is less precise in such cases, as observed in insight 2. As a result, non-incremental RS performs poorly compared to IRS in these situations, leading to a greater speedup with IRS.

## A.8    Threshold Parameter $\gamma$

Table 9 presents the proportion of cases for which $\underline{p_A} > \gamma$ for the $\gamma$ chosen through hyperparameter search in Section 5.4 for different $\sigma$ and networks.

Table 9: Proportion of $\underline{p_A} > \gamma$ for different $\sigma$ and networks.

| Dataset | Architecture | $\gamma$ | $\sigma$ | $\underline{p_A} > \gamma$ |
|---------|--------------|----------|----------|----------------------------|
| CIFAR10 | ResNet-20 | 0.99 | 0.25 | 0.346 |
|         |           |      | 0.5  | 0.162 |
|         |           |      | 1.0  | 0.034 |
| CIFAR10 | ResNet-110 | 0.99 | 0.25 | 0.362 |
|         |            |      | 0.5  | 0.146 |
|         |            |      | 1.0  | 0.034 |
| ImageNet | ResNet-50 | 0.995 | 0.5 | 0.292 |
|          |           |       | 1.0 | 0.14  |
|          |           |       | 2.0 | 0.04  |

For CIFAR10 ResNet-20, we observe that $\underline{p_A} > \gamma = 0.346$ when $\sigma = 0.25$ and $\underline{p_A} > \gamma = 0.034$ when $\sigma = 1.0$. Additionally, for ImageNet ResNet-50, the results show $\underline{p_A} > \gamma = 0.292$ when $\sigma = 0.50$ and $\underline{p_A} > \gamma = 0.04$ when $\sigma = 2.0$. As shown in Section 5, certifying larger $\sigma$ yields on average smaller $\underline{p_A}$. Expectedly, we see a smaller proportion of $\underline{p_A} > \gamma$ for larger $\sigma$ and vice versa.

## A.9 QUANTIZATION PLOTS

In this section, we present the ACR vs. time plots for all the quantization experiments. We use $n = 10^4$ for samples for certification of $g$. For certifying $g^p$, we consider $n_p$ values from $\{1\%, \dots 10\%\}$ of $n$. Note that these smaller values of $n, n_p$ compared to Section 5.1 allow us to perform a large number of experiments.

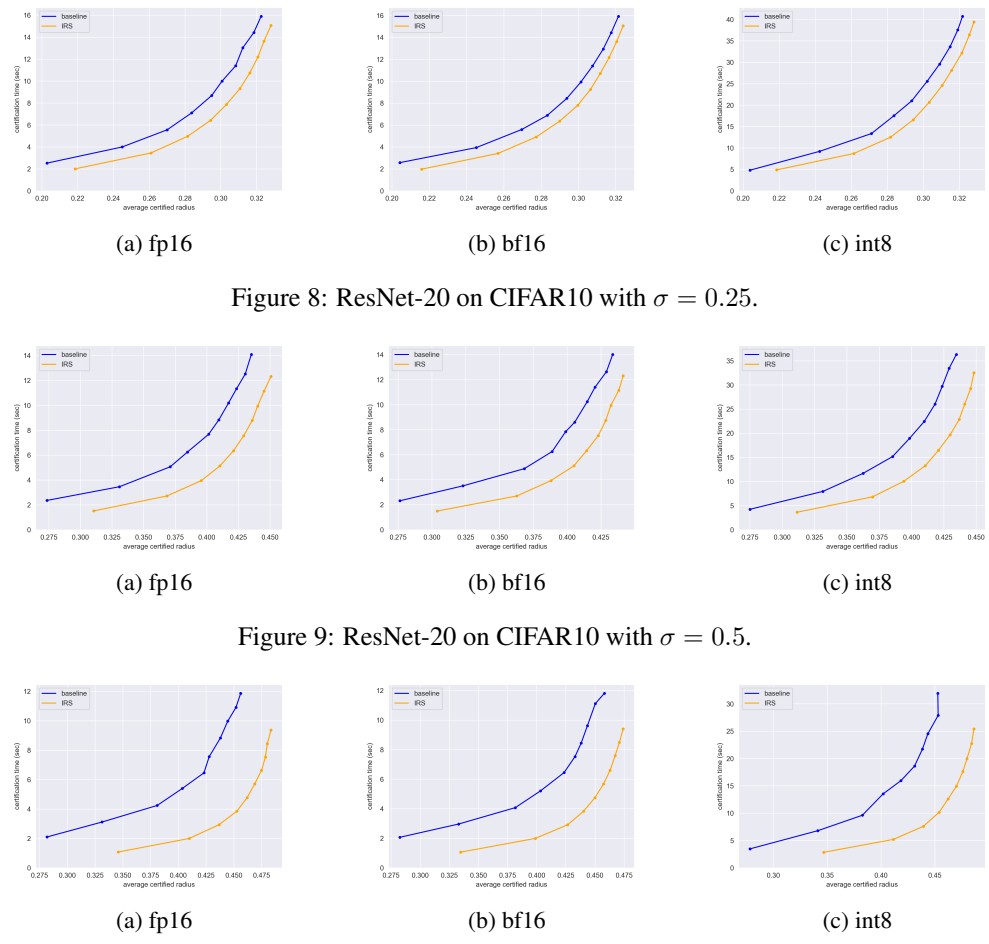

(a) fp16      (b) bf16      (c) int8

Figure 8: ResNet-20 on CIFAR10 with $\sigma = 0.25$.

(a) fp16      (b) bf16      (c) int8

Figure 9: ResNet-20 on CIFAR10 with $\sigma = 0.5$.

(a) fp16      (b) bf16      (c) int8

Figure 10: ResNet-20 on CIFAR10 with $\sigma = 1.0$.

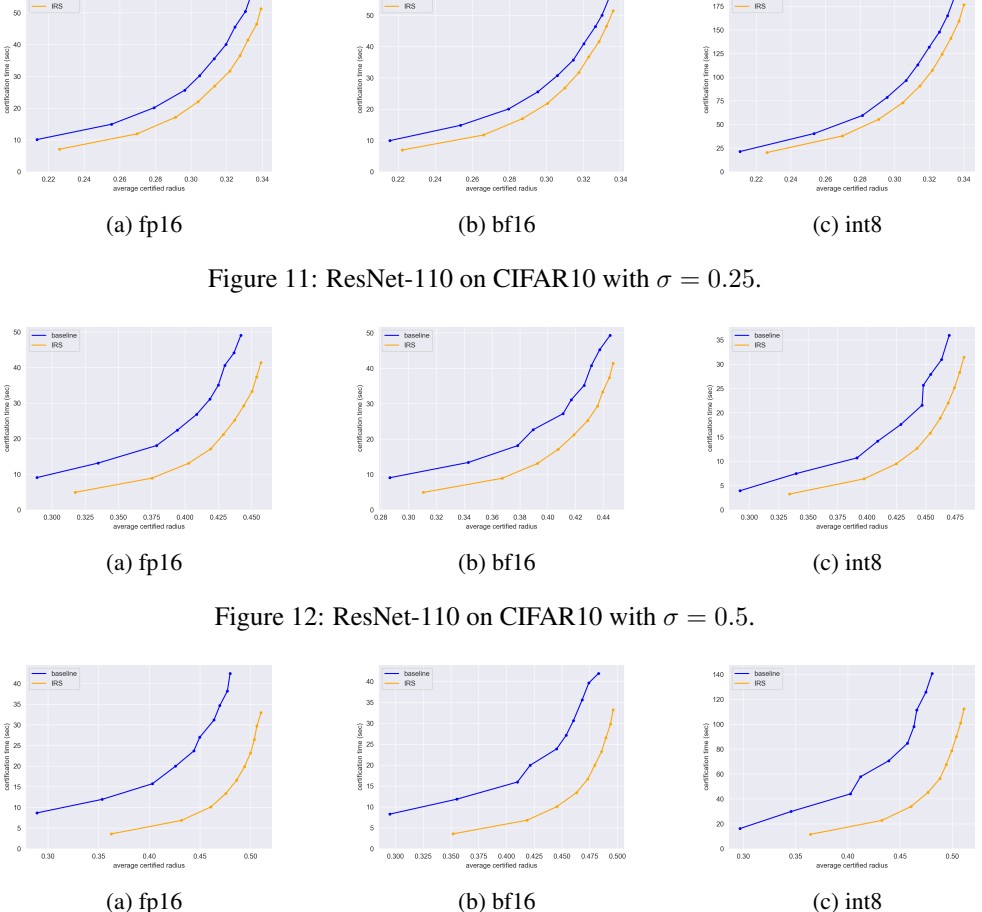

Figure 11: ResNet-110 on CIFAR10 with $\sigma = 0.25$.

Figure 12: ResNet-110 on CIFAR10 with $\sigma = 0.5$.

Figure 13: ResNet-110 on CIFAR10 with $\sigma = 1.0$.

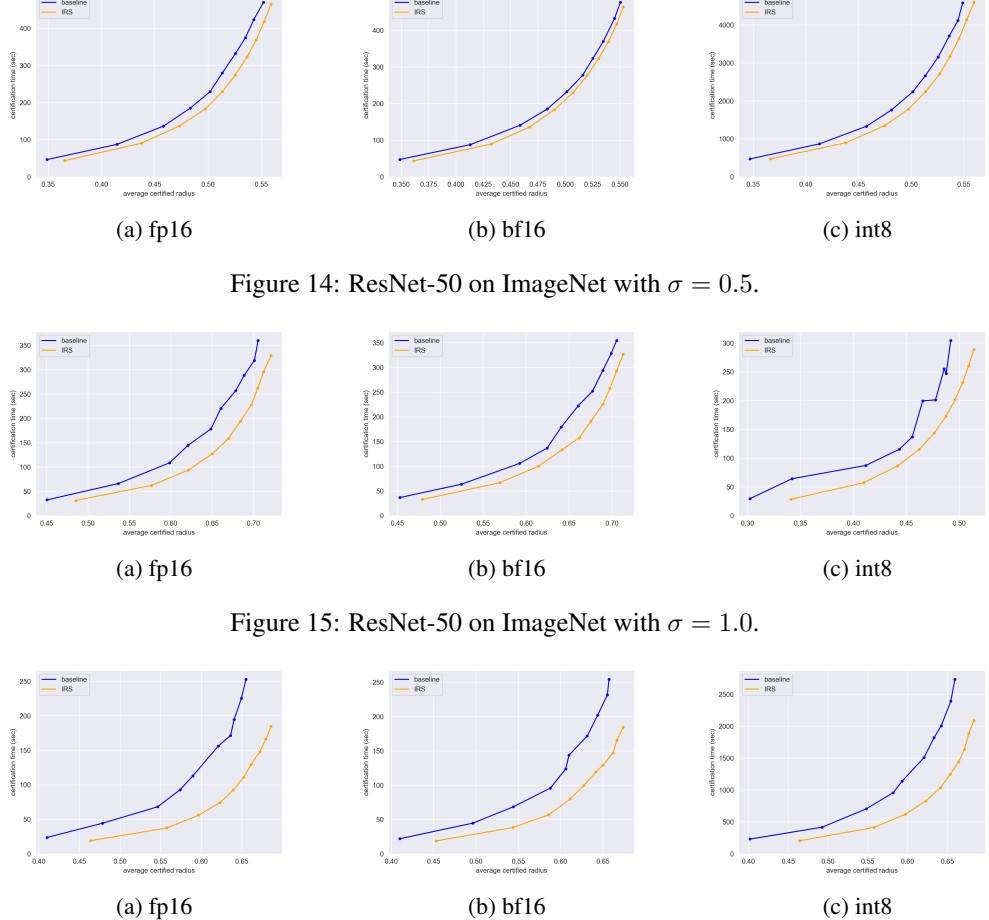

Figure 14: ResNet-50 on ImageNet with $\sigma = 0.5$.

Figure 15: ResNet-50 on ImageNet with $\sigma = 1.0$.

Figure 16: ResNet-50 on ImageNet with $\sigma = 2.0$.

