# OpenReview forum: "Incremental Randomized Smoothing Certification"
_ICLR.cc/2024/Conference — ICLR 2024 poster_

### Official Review · Reviewer_qzcN · 2023-10-25

**Soundness:** 3 good
**Presentation:** 4 excellent
**Contribution:** 3 good
**Rating:** 6
**Confidence:** 5

**Summary:**

This paper study certified robustness with randomized smoothing. The authors present a method that decreases the sample complexity of randomized smoothing in the setting where there is a classifier $f$ and an approximation of the same classifier $f^p$ (for example, $f^p$ is a quantized version of $f$). The method, called Incremental Randomized Smoothing, proposes to compute the certification of $f^p$ via the certificate of $f$. The method relies on estimating the disparity $\zeta_x$ which is the upper bound on the probability that outputs of $f$ and $f^p$ are distinct.

**Strengths:**

- Randomized smoothing is an important method, and currently the state-of-the-art approach, for certified robustness. Given the computational cost of this method, it is important to investigate how to make randomized more efficient. This paper investigates how to reduce the number of samples necessary for computing the certificate via Monte Carlo sampling.
- The paper is well-written, the theorems and algorithm are clear.

**Weaknesses:**

**Main Comment**.
I don't understand the main premise and setting used in this paper. I find one of the assumptions very strong and the practical implications of the method very limited. More detail below.

The authors state the following sentence in the abstract:

_``[...] when the smoothed model is modified (e.g., quantized or pruned), certification guarantees may not hold for the modified DNN, and recertifying from scratch can be prohibitively expensive.``_

Let $f$ be a base classifier, $f^p$ be a quantized version of $f$, and let $g$ be the smooth version of $f$ and $g^p$ be the smooth version of $f^p$.
It is true that a certificate computed from the _base model_ $f$ will not hold for the quantized version $f^p$. However, it would be possible to apply randomized smoothing directly to the quantized version $f^p$ via:
$$
g^p(x) = \underset{c \in \mathcal{Y}}{\operatorname{argmax}} \ \mathbb{P}_\epsilon [\ f^p(x + \epsilon) = c\ ]
$$

Instead, the authors propose to compute the certificate of $f^p$ by first computing the certificate for $f$ (which is the unquantized model and therefore expensive to run) and then computing the disparity $\zeta_x$, which is an upper bound on the probability that the outputs of $f$ and $f^p$ are different.
$\rightarrow$ It seems to me that this method is more expensive than computing the certificate directly on the quantized version $f^p$.

To claim that the approach is more efficient, the authors **assume** that the certificates for $f$ are available **for all $x$**, and therefore only the disparity $\zeta_x$ is needed to compute the new certificate. The authors state:

_``The IRS algorithm utilizes a cache $C_f$, which stores information obtained from the RS execution of the classifier $f$ for each input $x$. The cached information is crucial for the operation of IRS. $C_f$ stores the top predicted class index and its lower confidence bound $\underline{p_A}$ for $f$ on input $x$.``_

$\rightarrow$ The authors assume that the test data is already available and that the certificates have already been computed. I don't see how this can be realistic, especially since the authors mention that the quantized version $f^p$ can be used on edge devices, except perhaps if the model is only used for a limited set of inputs that are known in advance.

Can the authors comment on this and provide a practical use case that I may have missed?

**Questions:**

- Why the authors use the same Gaussian samples (same seed) in Algorithm 3? Is there any benefit?

---

> ### Author Response · Authors · 2023-11-14
>
> We thank the reviewer for insightful and constructive feedback.
>
> > It seems to me that this method is more expensive than computing the certificate directly on the quantized version.
> > To claim that the approach is more efficient, the authors assume that the certificates for
>  are available for all $x$
> > The authors assume that the test data is already available and that the certificates have already been computed. I don't see how this can be realistic.
>
> I think there is a misunderstanding with the motivation of our setup. Please see common response C1 for the motivation of the setup that we focus on in this work.
>
> IRS is not applicable while certifying individual approximate networks. We focus on the scenarios where we intend to certify multiple similar networks on a fixed test set to compute the certified radii. Computing certified radii for similar networks is a common occurrence when a user is comparing and selecting the best approximation for a network using techniques such as approximation tuning. For certifying the first network we cannot use IRS, however, for all subsequent network certifications, we can use IRS for faster certification.
>
>
> > Why do the authors use the same Gaussian samples (same seed) in Algorithm 3? Is there any benefit?
>
> Yes, using the same seed is crucial for making our algorithm 2x faster. The estimation of $\zeta_{x}$ involves computing $f(x+\epsilon)$ and $f^p(x+\epsilon)$ on Gaussian-corrupted versions of the input $x$. Given that we have already conducted this computation for $f$, we store the results for those samples in the cache. Consequently, we only need to compute $f^p$ on Gaussian-corrupted inputs. To ensure the soundness of this process, we utilize the same Gaussian corruptions for $f^p$ by retaining and reusing the seed. This approach incurs small memory and computational overhead.
>
> Thanks for pointing it out, we will further clarify our reasoning for using the same Gaussian samples in the paper.

---

> > ### Comment · Reviewer_qzcN · 2023-11-14
> > **Response to Rebuttal**
> >
> > Thank you for providing a detailed explanation of the motivation and setup. The addition of various scenarios to the paper would strengthen it and improve reader comprehension.
> > The method is interesting and useful in these specific contexts. I lean toward acceptance and increase the score accordingly.

---

> > > ### Author Response · Authors · 2023-11-14
> > >
> > > Thanks for your positive feedback and for increasing your score. We will add your suggestions to strengthen the paper in the revised version.

---

### Official Review · Reviewer_6uEd · 2023-10-31

**Soundness:** 3 good
**Presentation:** 4 excellent
**Contribution:** 2 fair
**Rating:** 6
**Confidence:** 3

**Summary:**

The paper deals with the problem of providing randomized smoothing-based certificates for modified neural networks. Given a modified version of a base model $f_p$ for some original base model $f$, the task is to provide a robustness certificate for the prediction of smoothed model $g_p$ at a point $x$ by efficiently reusing the values observed when calculating the certificate for smoothed model $g$ at the same $x$. The authors propose to do this using the fact that the difference between the value $g(x + \epsilon)$ and $g_p(x + \epsilon)$ around any point $x$ is very small (close to $0$) and the fact that the number of binomial samples required to estimate a parameter close to 0 is much smaller than the number of samples needed to estimate a binomial parameter close to 0.5. Given the difference in the value of $g$ and $g_p$ around a point $x$ and the certificate for $g(x)$, the authors give a formula to bound the certificate for $g_p$ at $x$.

**Strengths:**

- The idea of reusing the observations for calculating the certificate for $g$ to calculate the certificate of $g_p$ is novel and interesting.
- The authors also use a great insight that it is more efficient to estimate binomial parameters at extreme ends than near the middle.
- The paper is well-written and easy to understand.

**Weaknesses:**

- The practical usefulness of the proposed method is not clear. As randomized smoothing produces certificates at inference time, in order to calculate the certificate around a given point in this approach, the edge device would need access to both the original as well as the modified neural network models, which is not feasible.

**Questions:**

Please refer to the weaknesses section for questions.

---

> ### Author Response · Authors · 2023-11-14
>
> We thank the reviewer for insightful and constructive comments.
>
> > The practical usefulness of the proposed method is not clear. As randomized smoothing produces certificates at inference time, in order to calculate the certificate around a given point in this approach, the edge device would need access to both the original as well as the modified neural network models, which is not feasible.
>
> Please see common response C1 for motivation. IRS is not applicable for improving the inference time efficiency of RS. We focus on the common occurrences where the goal is to compute certified radius for multiple similar networks offline for comparison. For instance, when the user is selecting the best approximation for a network using the approximation tuning techniques as described in C1. We are happy to provide additional clarification to ensure a better understanding of our motivation.

---

### Official Review · Reviewer_VbzH · 2023-11-01

**Soundness:** 2 fair
**Presentation:** 3 good
**Contribution:** 2 fair
**Rating:** 6
**Confidence:** 3

**Summary:**

This work proposed how to certify a similar neural network via randomized smoothing by re-using the certification result from the original neural network. An IRS certification algorithm is provided in Algorithm 2 and its theory is provided in Theorem 2. The experiments on Cifar10 and ImageNet dataset showed the efficiency of the proposed algorithm for certifying different quantization models (fp16, bf16 and int8) and pruned models.

**Strengths:**

1. This work proposed a first incremental approach for randomized smoothing to certify a similar (compressed) version of the original neural network with improved efficiency by re-using the certification results.
2. The experiments results seem to be promising.

**Weaknesses:**

1. Demanding prerequisite: I am not sure how likely IRS algorithm is applicable in practice. It seems like IRS will require many prerequisite. For example, IRS needs to know the certification cache from the original neural network, which makes the requirement more demanding. If there is no such information, regular RS is still needed. As another requirement, IRS needs the modified network to be a good approximation of the original neural network. Otherwise, the accuracy might be reduced per theorem 2.
2. Novelty issue with the theory: for the theory part, most of the theorems are built upon theorem 1 in [Cohen et al 2019] and are direct application of that theorem, hence raising a novelty issue.

**Questions:**

See weakness above.

---

> ### Author Response · Authors · 2023-11-14
>
> We thank the reviewer for insightful and constructive feedback.
>
> > Demanding prerequisite: I am not sure how likely the IRS algorithm is applicable in practice. It seems like the IRS will require many prerequisites. For example, IRS needs to know the certification cache from the original neural network, which makes the requirement more demanding.
>
> Please see common response C1 for motivation. The prerequisite is common in the scenario where the objective is to compute a certified radius for multiple similar networks offline. For example, when the user aims to select the best approximation for a network by utilizing techniques such as approximation tuning.
>
> > Novelty issue with the theory: for the theory part, most of the theorems are built upon theorem 1 in [Cohen et al 2019] and are direct applications of that theorem, hence raising a novelty issue.
>
>
> Theorems 2, 3, and 4 combined are crucial to prove the soundness of our Algorithm. Theorem 1 by [Cohen et al 2019] considers standard RS for a single network. Our theorems show how to use the estimated value of $\zeta_x$ to transfer the certification guarantees across networks. We believe that this result is non-trivial and a novel theoretical contribution.

---

> > ### Comment · Reviewer_VbzH · 2023-11-19
> >
> > Thanks for your response. I have read the authors' reply and would like to keep my rating mainly due to weakness 1.

---

### Official Review · Reviewer_7muz · 2023-11-01

**Soundness:** 3 good
**Presentation:** 3 good
**Contribution:** 3 good
**Rating:** 6
**Confidence:** 3

**Summary:**

This work studies the efficiency of robustness certification in the case of approximated models by reusing the robustness guarantees in the original models. Specifically, the disparity between the original smoothed classifier and the approximated smoothed one is estimated to speed up the whole certification as it is relatively close to 0. The experiments show that the speed-up is obvious on different datasets with different models and smoothing parameters.

**Strengths:**

- The paper is well-written and easy to follow. The motivation is clear and important.
- The methodology is sound and it is friendly to read although it can be formally expressed with more complicated notations.
- The experiment is extensive and validates the effectiveness and efficiency of the method.

**Weaknesses:**

- Insight 1 in Section 3.1 is not very convincing in the sense of a single setting of n=1k and $\sigma=1$, where it usually costs 10k-100k samples for Monte Carlo sampling in estimation. More examples can be given to show $\zeta$ is small.
- For insight 2 and Figure 2, although the needed samples are much less compared to 0.5, it still needs 41.5k and there is no significant reduction compred to naive Monte Carlo randomized smoothing (10k-100k). A better way is to use an example of current estimation of $p_A$ and to show the needed samples are much less when estimating $\zeta$ compared to $p_A$.
- The choice of threshold $\gamma$ seems to be critical from the experiment results and the authors use grid search to optimize it. If I understand it correctly, whether to estimate $\zeta$ actually depends on whether $\zeta$ is closer to 0 than $p_A$ is to 1. So ideally, there can be some theoretical analysis for choosing $\gamma$ in terms of $\zeta$ and $p_A$.
- I think there is a missing ablation study of directly using naive Monte Carlo to estimate $\zeta$ instead of reusing seeds in terms of both certified radius and certification time.
- There are some typos and text messed up in the last two paragraphs in Section 3.2, e.g. In case, ... are correct with...

**Questions:**

See Weakness

---

> ### Author Response · Authors · 2023-11-14
>
> We thank the reviewer for insightful and constructive comments.
>
> > Insight 1 in Section 3.1 is not very convincing in the sense of a single setting of $n=1k$ and, where it usually costs $10k-100k$ samples for Monte Carlo sampling in estimation. More examples can be given to show is $\zeta_{x}$ is small.
>
> In Appendix A4, we show $\zeta_{x}$ for all networks and $\sigma$ values. We will point to this section in our insight 1. By considering $n=1000$ (smaller than $n=10k,100k$) we show that $\zeta_{x}$ is small with high confidence. The choice of smaller value $n$ for these experiments gives a strong argument that the IRS algorithm that uses the $\zeta_{x}$ estimate is likely to work well as it needs fewer samples to show that $\zeta_{x}$ is small. If the reviewer still thinks that experiments with larger values of $n$ would help strengthen this argument, we will add it to the paper.
>
>
> > For insight 2 and Figure 2, although the needed samples are much less compared to 0.5, it still needs 41.5k and there is no significant reduction compared to naive Monte Carlo randomized smoothing (10k-100k). A better way is to use an example of the current estimation of $p_A$ and to show the needed samples are much less when estimating $\zeta_{x}$ compared to $p_A$.
>
> Firstly, we appreciate your suggestion and we will make the change accordingly. We agree with the reviewer that the evidence on the distribution of $p_A$ strengthens the insight to motivate that the IRS algorithm will improve the sample efficiency. In appendix section A5, we show our observations on the distribution $p_A$ for different $\sigma$s. These observations show that $p_A$ is not close to $1$, especially when the $\sigma$ is larger. We will be happy to put these observations as one of our insights to make our presentation stronger.
>
>
> > The choice of threshold $\gamma$ seems to be critical from the experiment results and the authors use grid search to optimize it. If I understand it correctly, whether to estimate $\zeta_{x}$ actually depends on whether is closer $\zeta_{x}$ to 0 than $p_A$ is to 1. So ideally, there can be some theoretical analysis for choosing $\gamma$ in terms of $\zeta_{x}$ and $p_A$.
>
> Yes, your understanding is correct, in the IRS algorithm the decision to estimate $\zeta_{x}$ depends on whether $\zeta_{x}$  is closer to 0 than $p_A$ is to 1.
>
> While this analysis can be theoretically interesting, practically, we do not anticipate the analysis to help with the efficiency of the search compared to grid search for the following reasons. We found out that the final ACR is not too sensitive around the best choice of $\gamma$, and a simple grid search works quite well for this search. Since $\zeta_{x}$ is a function of each input $x$, for such an analysis to work it should rely on the distribution of $\zeta_{x}$ over different $x$s - computing that can be potentially expensive.
>
> > I think there is a missing ablation study of directly using naive Monte Carlo to estimate
>  instead of reusing seeds in terms of both certified radius and certification time.
>
> In theory, using naive Monte Carlo without seed reuse will not change the certified radius and it will be 2x slower than reusing seeds. We are happy to emphasize this in the paper. If the reviewer thinks that additional experiments will benefit the paper, we will add it in the revised version.

---

### Author Response · Authors · 2023-11-14
**Common Response**

Dear Area Chair and Reviewers,

We thank the reviewers for their constructive comments. We want to clarify the common point raised by the reviewers

**C1: Motivation of IRS (reviewers qzcN, 6uEd)**

The main motivation for IRS is the test phase when the user seeks to select an approximate (pruned, quantized) network from a range of possible networks with different approximations. This situation is encountered in numerous applications, making our approach valuable in improving the efficiency of model evaluation and comparison:

a) In the context of approximation tuning, there are multiple choices for approximation where different quantization or pruning ratio is applied at different layers. For instance, tools such as [1, 2, 3] use approximations iteratively and test the network at each step. The tuning should ensure a minimum user-defined metric such as certified accuracy. This certified accuracy is computed using RS on test data.

In these tools, performing RS to compute certified accuracy from scratch can take hours as shown in our experiments even for a single network (with only 500 test images). Consequently, the tuning process tends to be significantly time-consuming, especially given the multitude of network options. Existing approaches require running RS from scratch for each new network. In such instances, IRS emerges as a valuable time-saving alternative. Notably, IRS achieves significant time savings, over 13 hours on ResNet-50-ImageNet, and up to 4.1 times faster certification on ResNet-110-CIFAR10 reducing certification time from 2 hours 55 mins to 42 mins.

b) Often, a vendor releases an original model, and different users of the model have the option to either utilize the original model as-is or approximate (quantize, prune) it based on their specific device and application needs. In these situations, the vendor can provide certification for the original model along with the certification cache. The users may wish to compute the certified accuracy of their customized models  because the certification guarantee of the original model doesn’t hold for these approximate models [4][5]. Here, IRS can be useful, enabling faster certification for these customized models.

We will modify our introduction to make this motivation more explicit.



[1] Tianqi Chen, Thierry Moreau, Ziheng Jiang, Lianmin Zheng, Eddie Yan, Haichen Shen, Meghan Cowan, Leyuan Wang, Yuwei Hu, Luis Ceze, et al. 2018. “TVM: An automated end-to-end optimizing compiler for deep learning”, OSDI 2018

[2] Tianqi Chen, Lianmin Zheng, Eddie Yan, Ziheng Jiang, Thierry Moreau, Luis Ceze, Carlos Guestrin, and Arvind Krishnamurthy. 2018. “Learning to optimize tensor programs.” NeurIPS 2018

[3] Yifan Zhao, Hashim Sharif, Peter Pao-Huang, Vatsin Shah, Arun Narenthiran Sivakumar, Mateus Valverde Gasparino, Abdulrahman Mahmoud, Nathan Zhao, Sarita Adve, Girish Chowdhary, Sasa Misailovic, Vikram Adve “ApproxCaliper: A Programmable Framework for Application-aware Neural Network Optimization”. MLSys 2023

[4] Haowen Lin, Jian Lou, Li Xiong, and Cyrus Shahabi. “Integer-arithmetic-only certified robustness for quantized neural networks”, ICCV 2021

[5] Vikash Sehwag, Shiqi Wang, Prateek Mittal, and Suman Jana. “Hydra: Pruning adversarially robust neural networks”, NeurIPS 2020

---

### Author Response · Authors · 2023-11-22

Dear Reviewers,

We wish to inform you that we have updated the revised version of our paper. We are grateful for your insightful and constructive feedback. We have made the following changes based on your comments:

1. Updated the introduction and conclusion to make our motivation more explicit.
2. Updated our discussion on storing seeds, to show the advantage of storing seeds in comparison to naive Monte Carlo with fresh samples (page 6, paragraph 1) [Reviewer 7muz, qzcN]
3. Added more plots on $p_A$ distribution in the appendix to improve our presentation of insight 2 (page 19, Figure 6) [Reviewer 7muz]
4. Moved ablation study for $n$ to Appendix A.5 (page 17)

We are happy to incorporate any additional updates recommended by the reviewers.

---

### Meta-Review · Area_Chair_ejPa · 2023-12-14

**Metareview:**

The paper studied an interesting setting of robustness certification through randomized smoothing -- given a certification of a model f, how to certify another model f' which is similar but slightly different to the f (e.g., a compressed model). Reviewers unanimously agree that the proposed approach is both interesting and sound, offering a promising solution for this scenario. The analysis is rigorous, and the experiments effectively demonstrate the method's capabilities. While the immediate applicability of the "incremental randomized smoothing" setting might be limited, the paper's theoretical framework provides valuable insights, and thus we recommend for acceptance.

**Justification For Why Not Higher Score:**

The applications of IRS seem to be somehow narrow.

**Justification For Why Not Lower Score:**

The theoretical contributions and empirical evaluations are sufficient.

---

### Decision · Program_Chairs · 2024-01-16

Accept (poster)